# REViT: Roto-reflection Equivariant Convolutional Vision Transformer

**Sheir A. Zaheer** [1]   **Alexander C. Holston** [1]   **Chan Y. Park** [1]

## Abstract

In this paper, we propose a discrete roto-reflection group equivariant vision transformer with convolutional attention. Roto-reflection equivariant networks preserve the rotational, flip and positional symmetry in feature maps, making them useful for tasks where orientation of the inputs is relevant to the model outputs. In image classification and object detection, most of the studies on roto-reflection equivariant models have focused on using convolutional neural networks rather than vision transformers. In this paper, we examine the challenges involved in achieving equivariance in vision transformers, and we propose a simpler way to implement a discretized roto-reflection group equivariant vision transformer. The experimental results demonstrate that our approach outperforms the existing approaches for developing discrete roto-reflection group equivariant neural networks for image classification.

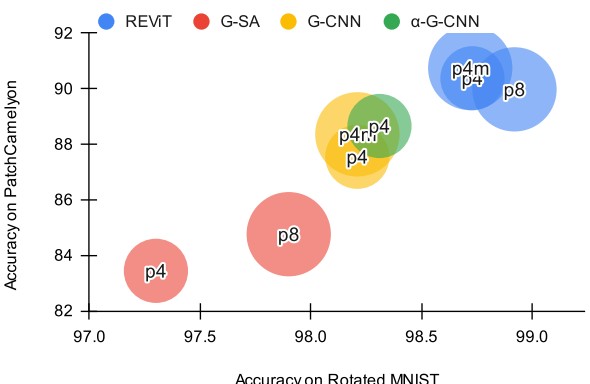

*Figure 1.* Rotation MNIST ($x$-axis) and PatchCamelyon ($y$-axis) performance of REViT vs existing approaches for discrete roto-translation and roto-reflection group equivariance (G-SA (Romero & Cordonnier, 2021), G-CNN (Cohen & Welling, 2016), $\alpha$-G-CNN (Romero et al., 2020)). Sizes of the bubbles are proportional to the number of elements (group order) in the rotation or roto-reflection ($p4m$) groups.

## 1. Introduction

Equivariant neural networks are a class of neural networks that maintain symmetry properties between their input and output representations (Guttenberg et al., 2016; Lim & Nelson, 2022; Wang et al., 2023). To propagate symmetry throughout the model, the equivariant neural networks are designed to ensure that all components of the model change predictably with respect to the transformations applied to the input. For example, in the case of rotation (or roto-reflection) equivariant models, if the input is rotated, the output of the network, along with all the intermediate feature maps, will also be rotated accordingly, preserving the underlying rotational symmetry (Cohen & Welling, 2016; Bekkers et al., 2018; Wiersma et al., 2020).

In traditional neural networks, the representation of an ob-

ject can change drastically when the object is rotated. However, in many applications, maintaining rotational symmetry of the representation is essential; e.g. understanding underlying patterns in molecular structure analysis (Liao & Smidt, 2022; Yi et al., 2023; Liao et al., 2023), digital pathology in medicine (Marcos et al., 2017; Veeling et al., 2018), or planning and navigation in robotics (Zhao et al., 2023; 2024). In computer vision, specifically, the preservation of roto-reflection symmetries is vital to ensure robustness to object orientations (Han et al., 2021; Lee et al., 2022) and to enable generalization across diverse datasets (Cohen & Welling, 2016; Romero & Cordonnier, 2021).

To achieve rotational and/or reflection equivariance, such networks typically employ specific architectural choices and operations. For example, they may use rotational convolutions (Marcos et al., 2016) or equivariant pooling layers and group convolutions (Cohen & Welling, 2016) to maintain symmetry properties throughout the network.

In 3D applications, equivariant networks have been well studied for tasks where instances of different classes have rotational symmetry; e.g., molecular structures (Schütt et al., 2021; Yi et al., 2023), or directional information, e.g., orientation of objects in 3D point clouds, need to be preserved

[1]KC Machine Learning Lab, Seoul, Rep. of Korea. Correspondence to: Sheir A. Zaheer <sheir@kc-ml2.com>, Chan Y. Park <chan.y.park@kc-ml2.com>.

(Thomas et al., 2018; Dym & Maron, 2020; Chen et al., 2021; Lee & Cho, 2024). Ideas inspired by attention mechanisms in graph neural networks (GNN) (Liao & Smidt, 2022; Liao et al., 2023) and vector neurons (Deng et al., 2021; Assaad et al., 2022) have been shown to be very effective for such problems. However, in all such applications, the input to the networks is 3D data with structural and/or positional information in a (Euclidean) space. However, that is not the case for 2D images/videos that are typically used in computer vision. These images are 2D projections of 3D objects/scenes, and the image pixel coordinates only describe the positions relative to an origin on the 2D projection. In spite of this, the orientation information with respect to the axis perpendicular to the projection is still preserved, and thus the symmetry information can be leveraged by a 2D rotation equivariant model.

Cohen & Welling (2016) showed how we can utilize the concept of rotational groups implemented through group convolutional neural networks (G-CNN) to implement models that generate rotation/roto-reflection equivariant feature maps for 2D images. This idea of group equivariant networks has been a building block for recent works on equivariant networks for oriented object detection (Han et al., 2021). Furthermore, as shown in (Cohen & Welling, 2016; Romero et al., 2020), the rotation equivariance realized through group convolutions can easily be extended to equivariance to other transformations like roto-translation and reflection by exploiting the inherent translation equivariance of convolutional neural networks (Bronstein et al., 2017).

In the case of vision transformers, however, such transformation equivariant designs are more complicated. Extending the same approach to design group equivariant self-attention is challenging because of the typical position encoding approaches utilized by vision transformers. However, by using self-attention mechanisms with relative position encoding (Shaw et al., 2018; Wu et al., 2021b), group equivariant self-attention can be developed (Romero & Cordonnier, 2021).

In this paper, we propose a vision transformer that achieves discrete roto-reflection equivariance with respect to the *E(2,N)* group, without relying on explicit positional encoding. We remove positional encodings by adopting convolutional patch embedding and convolutional self-attention mechanisms (Wu et al., 2021a). This design preserves spatial information within images while enabling generalization across different regions in a manner analogous to convolutional neural networks. At the same time, it retains the core advantages of vision transformers, namely self-attention and an improved ability to model global context (Dosovitskiy et al., 2021). Consequently, group self-attention can be implemented more simply as it no longer requires relative positional encoding. We evaluate our proposed approach by training roto-reflection equivariant vision transformers

(REViT) on different datasets. Our experimental results show that for the classification tasks tested, our approach outperforms both the group equivariant vision transformers with relative position encoding with significantly fewer parameters and the corresponding group equivariant convolutional neural networks (G-CNN) (Figure 1). Moreover, we also show that REViTs can be scaled to larger datasets such as ImageNet-1k (Deng et al., 2009).

The key contributions of this paper are:

- We propose discrete group convolutional self-attention (G-CSA) for roto-reflection (*E(2,N)*) equivariance without position encoding.

- We implement REViTs with G-CSA in their transformer blocks for multiple discrete roto-translation and roto-reflection group equivariant classification tasks on three different datasets.

- We also scale REViTs to ImageNet and contribute group equivariant vision transformers that can be used as roto-reflection equivariant backbones.[1]

## 2. Background

### 2.1. Groups and Group Equivariance

A *group* is formally defined as a pair $(G; \circ)$, where $G$ represents a set, and $\circ : G \times G \to G$ denotes a binary composition operation. All groups must satisfy four axioms:

1. **Closure:** This means that for any $g$ and $h$ belonging to $G$, the result of their composition $g \circ h$ also belongs to $G$.

2. **Associativity:** For all elements $f, g, h$ in $G$, the composition is associative, expressed as $f \circ (g \circ h) = (f \circ g) \circ h = f \circ g \circ h$.

3. **Identity:** There exists an element $e$ in $G$ such that for any $g$ in $G$, $e \circ g = g \circ e = g$.

4. **Inverses:** For every element $g$ in $G$, there exists an element $g^{-1}$ in $G$ such that $g^{-1} \circ g = g \circ g^{-1} = e$.

For symmetry-preserving groups, each element $g$ within $G$ corresponds to a specific symmetry transformation.

**Group Equivariance**   Let $\Phi : V_1 \to V_2$ be a map between two spaces $V_1$ and $V_2$, and let $\rho_1$ and $\rho_2$ be actions of a group $G$ on $V_1$ and $V_2$ respectively. Then, $\Phi$ is said to be $G$-equivariant if the following condition holds:

$$\Phi[\rho_1(g)f] = \rho_2(g)[\Phi[f]], \quad \forall g \in G, f \in V_1. \quad (1)$$

---

[1]Available at *https://github.com/kc-ml2/revit*

This means that applying a group action $\rho_1(g)$ to the input before applying the map $\Phi$ is equivalent to applying the corresponding action $\rho_2(g)$ to the output after applying $\Phi$. This property captures the idea that the map $\Phi$ respects the symmetries encoded in the group action.

## 2.2. Related Works

In recent years, different approaches for roto-reflection equivariant neural networks have been proposed for a diverse set of tasks (Marcos et al., 2016; Cohen & Welling, 2016; Romero et al., 2020; Cohen et al., 2021; Khan et al., 2022). This paper focuses on roto-reflection equivariant models for 2D computer vision, so in the following paragraphs, we cover approaches proposed for roto-reflection equivariant image feature extraction and classification.

**Rotational Convolutions and Equivariant Pooling Layers:** One of the fundamental architectural designs in rotation equivariant models is rotational convolutions (Marcos et al., 2016; Follmann & Bottger, 2018; Wei et al., 2023). These convolutions are designed to apply the same transformation to the input regardless of its orientation. This ensures that the output of the convolutional layer remains equivariant to rotations, preserving rotational symmetry in the learned representations. Equivariant pooling layers are another essential component in rotation equivariant models (Cohen & Welling, 2016; Lang & Weiler, 2020; Cohen et al., 2021). These layers aggregate features in a way that maintains equivariance to rotations.

**Group Equivariant Convolutions:** Group equivariant convolutions (Cohen & Welling, 2016) are a specialized form of convolutions in which the elements of the group are expanded into the depth dimension of the network. In rotation equivariant models, group convolutions are often used to enforce equivariance by ensuring that each grouping of input channels undergoes a rotational transformation corresponding to all elements of the rotational group. Group equivariant convolutions have also been extended towards steerable convolution models aimed at *SE(2)*-equivariance in the continuous domain (Sanborn et al., 2024).

**Equivariant Attention:** In transformer-based models, attention mechanisms play a crucial role in capturing relationships between different parts of the input (Han et al., 2022; Khan et al., 2022). To enforce equivariance in transformers, attention mechanisms need to be modified to operate on rotations in an equivariant manner to ensure that the learned attention weights remain consistent across different rotations of the input. Past works have done that using techniques like graph attention (Fuchs et al., 2020; Satorras et al., 2021) or self-attention with specifically formulated relative position encoding (Romero & Cordonnier, 2021).

Beyond discrete roto-reflection equivariant neural networks, a broad body of work has studied equivariant neural networks and group convolutions beyond discrete roto-reflection CNNs. Foundational works generalized convolution and equivariance to compact groups, homogeneous spaces, and Lie groups (Kondor & Trivedi, 2018; Cohen et al., 2019b; Finzi et al., 2020), while Harmonic Networks and gauge-equivariant CNNs explored rotation-equivariant and geometric formulations (Worrall et al., 2017; Cohen et al., 2019a). Equivariant learning has also been widely applied in 3D vision and scientific modeling through spherical, *SO(3)*, and *E(3)* equivariant architectures (Cohen et al., 2018; Esteves et al., 2018; Batzner et al., 2022).

## 3. Position Encoding and Equivariance in Transformers

In this section we discuss the self-attention mechanism that is used in typical transformer architecture. In particular, we delve into position encoding methodologies as we understand them to be the biggest hurdle in the development of ViTs (Dosovitskiy et al., 2021) that are both roto-reflection equivariant and simple to implement. We will elaborate on those challenges in Section 3.2, after an overview of self-attention and why (vision) transformers need position encoding.

### 3.1. Self-attention in Vision Transformers

Consider an input matrix $X \in \mathbb{R}^{N \times C_{\text{in}}}$ comprising $N$ tokens, each with $C_{\text{in}}$ dimensions. A self-attention layer transforms $X$ into an output matrix $Y \in \mathbb{R}^{N \times C_{\text{out}}}$ as follows:

$$Y = \text{SA}(X) := \text{softmax}\left(\frac{A}{\sqrt{d_k}}\right) X W_V, \qquad (2)$$

where $W_V \in \mathbb{R}^{C_{\text{in}} \times C_h}$ is the value matrix, $A \in \mathbb{R}^{N \times N}$ represents the attention scores matrix, and softmax $\left(\frac{\cdot}{\sqrt{d_k}}\right)$ computes the attention probabilities. $d_k$ is the dimensionality of the input embedding. The matrix $A$ is computed as:

$$A := X W_Q (X W_K)^\top, \qquad (3)$$

where $W_Q, W_K \in \mathbb{R}^{C_{\text{in}} \times C_h}$ are the query and key matrices, respectively. It has been observed to be beneficial to utilize multiple self-attention operations (or "heads") concurrently to enable diverse parts of the input to be attended to. In the multi-head self-attention formulation, the outputs of $H$ heads, each having output dimension $C_h$, are concatenated and projected onto $C_{\text{out}}$ as:

$$\text{MHSA}(X) := \text{concat}_{h \in [H]} \left[\text{SA}_h(X)\right] W_{\text{out}} + b_{\text{out}}, \quad (4)$$

where $W_{\text{out}} \in \mathbb{R}^{H C_h \times C_{\text{out}}}$ is the projection matrix and $b_{\text{out}} \in \mathbb{R}^{C_{\text{out}}}$ is the bias term.

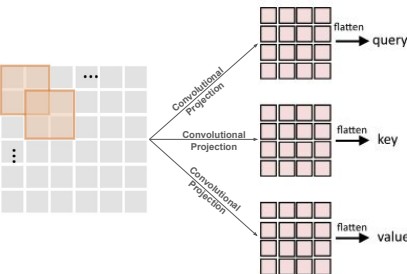

*Figure 2.* Convolutional Projection of Key, Query and Values (Wu et al., 2021a).

The self-attention operation defined in (2) and (4) is permutation equivariant. In simpler terms, if the rows of $X$ are rearranged (permuted), the resulting output $Y$ will also undergo the same permutation, maintaining the same relative order of the elements. This property means that self-attention ignores the input order, treating the inputs as a set rather than a fixed sequence. For example, when processing an image, self-attention treats pixels as an unordered set, ignoring their spatial arrangement and structure.

To address this limitation, position encodings are commonly added to the input representations. These position encodings enrich the input embeddings with information about the positions of elements in the set, allowing the model to distinguish between different elements based on their positions in the sequence.

### 3.2. Position Encoding's Effect on Equivariance and Complexity

Position encoding may influence both the equivariance properties and computational cost of self-attention networks. Absolute position encoding (Vaswani et al., 2017) assigns a unique vector to each position, causing the model to learn position-specific patterns and breaking equivariance to transformations such as translations or permutations In contrast, relative position encoding (RPE) (Shaw et al., 2018) encodes position differences, thus preserving translation equivariance similarly to convolutional networks. This idea can be extended to group-equivariant vision transformers (Romero & Cordonnier, 2021) by incorporating rotation group encodings $G_{e(j)-e(i)}$ alongside horizontal and vertical RPE terms $P_{x(j)-x(i)}$ and $P_{y(j)-y(i)}$, resulting in the following:

$$A := X_i W_Q ((X_j + P_{x(j)-x(i)} + P_{y(j)-y(i)} + G_{e(j)-e(i)}) W_K)^\top. \quad (5)$$

However, unlike absolute encodings, which are added once to the input, RPE must be computed in every attention step of every layer, leading to additional complexity.

### 3.3. ViT without Position Encoding

Wu et al. (2021a) introduced the Convolutional Vision Transformer (CvT), an architecture that enhanced ViTs by incorporating convolutional operations. This integration was achieved through two main innovations: a hierarchical transformer structure with a convolutional patch embedding, and a convolutional transformer block that uses convolutional projections (Figure 2). CvT outperformed other ViTs with fewer parameters and FLOPs (Wu et al., 2021a). Notably, CvT does not require position encoding, which is our main motivation to adopt convolutional self-attention in REViT. These changes bring the benefits of convolutional neural networks, such as shift equivariance (Bronstein et al., 2017), while retaining the strengths of transformers, including attention and global context.

Since the purpose of enriching the group self-attention with RPE (Romero & Cordonnier, 2021) is to achieve shift equivariance, convolutional patches and convolutional self-attention are key building blocks of our proposed REViTs without RPE.

## 4. REViT: Roto-reflection Equivariant Vision Transformer

In this work, we used a standard ViT architecture with a few modifications, as shown in Figure 3. Apart from removing position encoding, we also replace the typical self-attention with group convolutional self-attention and add a lifting layer prior to multi-head attention as a means to lift the input $X$ from the pixels in the Euclidean space to a feature space transformed by a discrete roto-translation group. The following subsections describe these two modules.

### 4.1. Lifting Layer

The lifting layer takes an input signal $f : \mathbb{R}^2 \to \mathbb{R}^C$ (e.g., an image with $C$ channels) and lifts it to a spatial location associated with multiple transformations under group $G$. In our case, the lifting layer transforms the 2D image input into a lifted feature map that depends both on the position $x \in \mathbb{R}^2$ and the orientation $\theta \in [0, 2\pi]$. The lifted feature map now lives in a three-dimensional position orientation space $\mathbb{R}^2 \times S^1$ (where $S^1$ is the orientation space).

In the lifting layer, we apply a rotational convolution (Bekkers et al., 2018) between the input image and a set of rotated filters:

$$[f * k](x, g) = \sum_c \int_{\mathbb{R}^2} f(x') k_c(g^{-1}(x' - x)) \, dx'. \quad (6)$$

This equation follows the formulation of a typical convolution, $(f * k)(x) = \sum_c \int_{\mathbb{R}^2} f(x') k_c(x' - x) dx'$, but instead of using a fixed kernel $k(x' - x)$, we apply a rotated version: $k_c(g^{-1}(x' - x))$ where $g$ is a rotational transformation. The

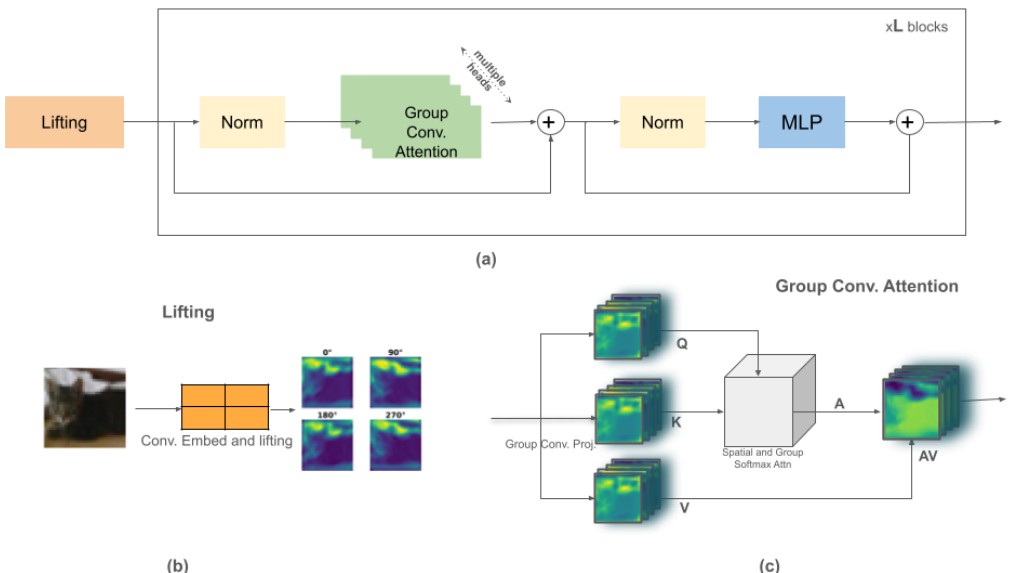

*Figure 3.* REViT: (a) $L$ transformer blocks preceded by a lifting layer, (b) illustration of lifting layer for a roto-reflection group with 4 elements, each rotated by 90°, and (c) 3D group convolutional self-attention with two dimensions for spatial projection and 1 dimension along the group elements from lifting layer

inverse rotation $g^{-1}$ maps the shifted coordinate $x' - x$ back to the kernel's canonical space. Additionally, $\sum_c$ is the summation over channels, i.e. RGB channels in color images or feature maps from a previous layer of a deep learning model.

Since this paper deals with discrete roto-translations, we adapt this lifting operation for discrete rotation groups, where rotations belong to a finite set $\Theta = \{\theta_1, \theta_2, \ldots, \theta_N\}$ and the input function is $f : \mathbb{Z}^2 \to \mathbb{R}^C$ (an image with discrete pixels with C channels). We define the lifting over position $x$ and the discrete orientations $\theta$ associated with the discrete rotation group, $\mathbb{C}_N$:

$$F(x, g) = [f * k](x, g) = \sum_c \sum_{x' \in \mathbb{Z}^2} f(x') k_c(g^{-1}(x' - x)), \quad (7)$$

where $F : \mathbb{Z}^2 \times \mathbb{C}_N \to \mathbb{R}^{C'}$ is the lifted feature map, defined over discrete positions and $N$ discrete orientations, $k_c(g^{-1}(x' - x))$ with $g = R_\theta$ is a discrete rotation-aware convolutional kernel defined for each $\theta_j = \frac{2\pi j}{N}$, where $j \in \{0, \ldots, N-1\}$. Figure 3(b) shows the lifted feature maps for a roto-translation group with four elements ($N = 4$). This group of four discrete rotations with an angular increment of $\pi/2$ is also known as a $p4$ group.

## 4.2. Group Convolutional self-attention

In this paper, we propose group convolutional self-attention (G-CSA) as the attention mechanism for MHSA modules in REViT. All REViT models that we developed for this paper utilize G-CSA (Figure. 3(c)). G-CSA is a mapping from functions defined on an affine group $G = \mathbb{Z}^2 \rtimes \mathbb{C}_N$ to functions on the same group $G$ modified by the action of group elements.

It operates on input functions $F : \mathbb{Z}^2 \times \mathbb{C}_N \to \mathbb{R}^{C'}$, where $F$ comes from the previous transformer block with G-CSA, except in the case of the first block, where it comes from the lifting layer. In the following subsections, we explain how we first obtain the *query-key-value* projections from $F$ and then calculate G-CSA.

### 4.2.1. QUERY-KEY-VALUE COMPUTATION

In standard convolutional self-attention (Wu et al., 2021a), self-attention learns relationships between spatial locations on an input $f : \mathbb{R}^2 \to \mathbb{R}^C$. In Group Equivariant Convolutional Self-Attention (G-CSA), we extend this concept to a lifted feature space, where each spatial location is associated with multiple transformations $g \in G$.

The *query*, *key*, and *value* mappings in this lifted space are computed using:

$$\begin{aligned} Q(x, g) &= W_Q * F(x, g), \quad K(x, g) = W_K * F(x, g), \\ V(x, g) &= W_V * F(x, g), \end{aligned}$$

$$\quad (8)$$

where $F$ represents the feature at position $x$ and transformation $g$. $W_Q, W_K, W_V$ are group-equivariant convolutional kernels, calculated by group equivariant convolutional projections of the input followed by a group equivariant convolution operation, and $*$ denotes group equivariant convolution.

As we use convolutional projections for both lifting and G-CSA rather than linear projections of the patches, the *key*, *query*, and *value* projections in G-CSA are implemented as three-dimensional convolutions (Figure 3(c)): two spatial dimensions that preserve 2D translation equivariance, intrinsic to convolutional layers (Bronstein et al., 2017) and a depth dimension for the group elements.

### 4.2.2. G-CSA COMPUTATION

To implement G-CSA, we modify typical self-attention, defined in (2), by incorporating group structure:

$$G\text{-}CSA(x,g) := \sum_{y \in \mathcal{N}(x)} \sum_{h \in G} A(x,g;y,h)V(y,h) \quad (9)$$

where $\mathcal{N}(x)$ denotes the local neighborhood of $x$ defined by the receptive field of the convolution, and the attention weights $A$ are calculated as:

$$A(x,g;y,h) := \frac{\exp\left(\frac{\langle Q(x,g),K(y,h)\rangle}{\sqrt{d_k}}\right)}{\sum_{y',h'} \exp\left(\frac{\langle Q(x,g),K(y',h')\rangle}{\sqrt{d_k}}\right)} \quad (10)$$

where $\langle \cdot, \cdot \rangle$ represents the dot product.

This ensures that attention operates over both spatial and group dimensions while preserving translation equivariance via convolution.

### 4.2.3. EQUIVARIANCE OF G-CSA

Here, we show that G-CSA is group equivariant. Let $T_g$ be a group transformation acting on a feature function $F : X \times G \to \mathbb{R}^d$. A transformation $g \in G$ acts as:

$$(T_gF)(x,h) = F(g^{-1}x, g^{-1}h) \quad (11)$$

where $g^{-1}x$ is undoing the transformation $g$ on the spatial point $x$, and $g^{-1}h$ refers to inverting the transformation $g$ before applying the transformation $h$.

Given the transformed feature $T_gF$, we compute the *query, key, and value* mappings for the transformed feature from (8) using group-equivariant convolutions:

$$Q_g(x,h) := W_Q * (T_gF)(x,h), \ K_g(x,h) := W_K * (T_gF)(x,h),$$
$$V_g(x,h) := W_V * (T_gF)(x,h) \quad (12)$$

Since these are implemented via equivariant convolutions, combining (12), (11), and (8) we get:

$$Q_g(x,h) = Q(g^{-1}x, g^{-1}h), \quad K_g(x,h) = K(g^{-1}x, g^{-1}h),$$
$$V_g(x,h) = V(g^{-1}x, g^{-1}h) \quad (13)$$

Then, we compute the attention weights for the transformed feature using (10):

$$A_g(x,h;y,h') := \frac{\exp\left(\frac{\langle Q_g(x,h),K_g(y,h')\rangle}{\sqrt{d_k}}\right)}{\sum_{y',h''} \exp\left(\frac{\langle Q_g(x,h),K_g(y',h'')\rangle}{\sqrt{d_k}}\right)} \quad (14)$$

Using the equivariance of $Q$ and $K$ from (13):

$$\langle Q_g(x,h), K_g(y,h')\rangle = \langle Q(g^{-1}x, g^{-1}h), K(g^{-1}y, g^{-1}h')\rangle. \quad (15)$$

Using (15) and the invariance of the dot product to the transformation applied to both vectors, we rewrite (14) as:

$$A_g(x,h;y,h') = A(g^{-1}x, g^{-1}h; g^{-1}y, g^{-1}h') \quad (16)$$

From (9), G-CSA output for the transformed feature is:

$$G\text{-}CSA_g(x,h) := \sum_{y,h'} A_g(x,h;y,h')V_g(y,h') \quad (17)$$

Substituting $A_g$ with (16) and the equivariance of $V_g$ from (13):

$$V_g(y,h') = V(g^{-1}y, g^{-1}h'), \quad (18)$$

we obtain:

$$G\text{-}CSA_g(x,h) = \sum_{y,h'} A(g^{-1}x, g^{-1}h; g^{-1}y, g^{-1}h')V(g^{-1}y, g^{-1}h'). \quad (19)$$

Rewriting with $y' = g^{-1}y$, $h'' = g^{-1}h'$ and using (9):

$$G\text{-}CSA_g(x,h) = \sum_{y',h''} A(g^{-1}x, g^{-1}h; y', h'')V(y', h'')$$
$$= G\text{-}CSA(g^{-1}x, g^{-1}h) \quad (20)$$

Since G-CSA is a feature function, using (11) we get:

$$G\text{-}CSA_g(x,h) = T_g\left(G\text{-}CSA(x,h)\right) \quad (21)$$

which shows *equivariance*:

$$G\text{-}CSA(T_gF) = T_g\left(G\text{-}CSA(F)\right) \quad (22)$$

## 5. Experimental Results

### 5.1. Experimental Setup

We validate the performance of REViT on three publicly available datasets typically used for testing roto-translation and/or roto-reflection equivariant models.

1. The **rotated MNIST** dataset (Larochelle et al., 2007) is a classification dataset commonly used as a benchmark to evaluate rotation equivariance. It contains 62,000 28x28 rotated images of handwritten digits.

*Table 1.* Classification accuracy (%) and number of parameters for each model on Rotated MNIST and PatchCamelyon datasets. † G-SA with RPE (Romero & Cordonnier, 2021)

| Approach | Model Config | Rotated MNIST | | PatchCamelyon | |
|---|---|---|---|---|---|
| | | Acc. (%) | Params | Acc. (%) | Params |
| G-SA† | Z2-SA | 96.37 | | 83.04 | |
| | p4-SA | 97.30 | 44.67K | 83.44 | 205.66K |
| | p8-SA | 97.90 | | 83.58 | |
| | p4m-SA | - | | 84.76 | |
| G-CSA | Z2-CSA | 95.97 | | 86.4 | |
| | p4-CSA | 97.48 | 44.28K | 90.38 | 94.35K |
| | p8-CSA | **98.03** | | 89.98 | |
| | p4m-CSA | - | | **90.75** | |

*Table 2.* Model complexity comparison of G-CSA with G-SA with RPE in terms of total multiplication and addition operations (in millions) and the peak model memory size (in Megabytes) when trained on batches of 16 28x28 images.

| Approach | Model | Mul-Add (M) | Total Size (MB) |
|---|---|---|---|
| G-SA with RPE | Z2-SA | 60.16 | 29.58 |
| | p4-SA | 232.49 | 161.77 |
| | P8-SA | 462.29 | 198.05 |
| G-CSA (Ours) | Z2-CSA | 29.10 | 9.09 |
| | p4-CSA | 116.37 | 35.84 |
| | p8-CSA | 232.98 | 71.72 |

2. **PatchCamelyon** (Veeling et al., 2018) contains 327,000 RGB images of breast tissue, each 96x96 in size and labeled tumorous or non-tumorous. These patches are extracted from the Camelyon16 dataset (Ehteshami Bejnordi et al., 2017) and labeled as tumorous if the central region includes any tumor pixel(s).

3. **CIFAR-10** is a benchmark image classification dataset containing 60,000 color images of size 32×32 across 10 object categories (Krizhevsky, 2009).

We have used the following group configurations to test the proposed group equivariant convolutional self-attention (G-CSA) for vision transformers:

**2D Integer Translation**. We use *Z2-CSA* models to test 2D integer translation ($\mathbb{Z}^2$) group equivariant classification performance.

**Roto-translation: *SE(2,N)* or *pN***. We use *pN-CSA* models for discrete roto-translation equivariance. The *pN* group has $N$ elements, each rotated by $\frac{360}{N}°$.

**Roto-reflection: *E(2,N)* or *pNm***. We use *pNm-CSA* models for discrete roto-reflection equivariance. The *pNm* group has $N \times 2$ elements: $N$ elements rotated by $\frac{360}{N}°$ and $N$ reflections. Occasionally, we also use the reflection-only group *Z2m* to compare with previous works that used reflection-only groups.

We compare *G-CSA* models against the corresponding models with group equivariant self-attention (G-SA) enriched with relative position encoding (Romero & Cordonnier, 2021) and group equivariant convolutional neural networks proposed in Cohen & Welling (2016) and Romero et al. (2020).

**5.2. Comparison against G-SA with RPE**

5.2.1. PERFORMANCE COMPARISON

Table 1 shows the performance comparisons of REViTs with group equivariant ViTs with RPE. The results show that our models match or outperform the models where group equivariant self-attention (G-SA) needed to be enriched with

RPE in every attention layer (Romero & Cordonnier, 2021).

5.2.2. COMPLEXITY COMPARISON

Here we compare the computational complexity of REViTs with our proposed G-CSA against the ViTs implemented with group self-attention with RPE. We trained similarly sized models (approximately 44K parameters) on Rotated MNIST with a batch size of 16 and noted the memory required for each model along with the total number of multiplication and addition operations required by each model. Table 2 summarizes the results of that comparison and shows a significant reduction in the number of operations in the case of REViTs with G-CSA.

**5.3. Performance Comparison with G-CNNs**

Table 3 shows a comparison between REViTs with G-CSA and group-equivariant CNNs across Rotated MNIST, CIFAR-10, and PatchCamelyon. Across all datasets and symmetry groups, REViTs yield consistent improvements over the corresponding equivariant G-CNN baselines. On Rotated MNIST, CSA improves performance for all rotation-equivariant models, with p8-CSA achieving the highest accuracy. On CIFAR-10, where gains from equivariance are generally more modest, G-CSA continues to provide stable but smaller improvements for both roto-translation and roto-reflection equivariant architectures. The largest relative improvements are observed on PatchCamelyon, where CSA-based equivariant models outperform all CNN baselines, including higher-order symmetry variants, while using fewer parameters. Overall, these results suggest that G-CSA improves group equivariant classification without relying on any significant increase in model sizes and complexities.

**5.4. Exploratory Analysis of G-CSA Configurations**

5.4.1. ABLATION - W/O EQUIVARIANT ATTENTION

To demonstrate the effectiveness of G-CSA in REViTs, we also conducted experimental comparisons with vanilla ViTs (Dosovitskiy et al., 2021) with and without roto-translation data augmentations. The results of this ablation are summarized in Table 4. The results show that though the augmenta-

*Table 3.* Performance Comparisons on (a) Rotated MNIST, (b) CIFAR-10, and (c) PatchCamelyon datasets with Group Equivariant CNNs (Cohen & Welling, 2016). †(Romero et al., 2020)

*(a)* Rotated MNIST

| Model | Acc. (%) | Params |
|---|---|---|
| Z2-CNN | 94.7 | 21.8K |
| p4-CNN | 98.21 | 77.8K |
| p8-CNN | 98.5 | 77.8K |
| α-p4-CNN† | 98.31 | 73.1K |
| Z2-CSA | 95.97 | 44.3K |
| p4-CSA | 98.73 | 97.7K |
| p8-CSA | **98.92** | 101.4K |

*(b)* CIFAR-10

| Model | Acc. (%) | Params |
|---|---|---|
| Z2-CNN | 90.56 | |
| Z2m-CNN | 91.16 | 1.4M |
| p4m-CNN | 92.47 | |
| Z2-CSA | 89.2 | |
| Z2m-CSA | 91.18 | 1.12M |
| p4m-CSA | **92.68** | |

*(c)* PatchCamelyon

| Model | Acc. (%) | Params |
|---|---|---|
| Z2-CNN | 84.07 | 130.60K |
| p4-CNN | 87.55 | 129.65K |
| p4m-CNN | 88.36 | 124.21K |
| $\alpha_F$-p4-CNN† | 88.66 | 140.45K |
| $\alpha_F$-p4m-CNN† | 89.12 | 141.22K |
| Z2-CSA | 86.4 | |
| p4-CSA | 90.38 | 94.35K |
| p4m-CSA | **90.75** | |

*Table 4.* Rotated MNIST performance Comparison of REViTs with G-CSA against vanilla ViTs with typical self-attention (SA) of similar size and trained for 200 epochs. ViTs were trained with and without data augmentations of discrete random 45° rotations and translations.

| Model | Attention | Results |
|---|---|---|
| ViT w/o data augmentation | SA | 81.25 |
| ViT w/ data augmentation | SA | 91.67 |
| CvT w/ data augmentation | CSA | 91.46 |
| REViT | $Z2CSA$ | 95.97 |
| REViT | $p8CSA$ | 98.03 |

*Table 5.* Effects of G-CSA's group order and kernel size

*(a)* Group order vs. Accuracy ($5x5$ convolutions)

| Group | Acc.(%) |
|---|---|
| $z2$ | 95.97 |
| $p4$ | 98.73 |
| $p8$ | 98.92 |
| $p12$ | **99.01** |
| $p16$ | 98.89 |

*(b)* Convolutional Kernels vs. Accuracy ($p4 - CSA$)

| Kernel | Acc.(%) |
|---|---|
| 3×3 | 97.98 |
| 5×5 | **98.73** |
| 7×7 | 98.65 |
| 9×9 | 98.53 |
| 11×11 | 97.39 |

tions have a significant impact on making models invariant to transformations in data, they still do not perform as well as equivariant models when the model and dataset sizes are controlled for.

### 5.4.2. ROTATION GROUP ORDER

Table 5a lists how different rotation group orders affect the accuracy of REViT on Rotated MNIST dataset. We can see that increasing the elements of the rotation group does not always increase performance.

### 5.4.3. G-CSA KERNEL SIZES

Since REViT uses G-CSA, the choice of convolutional kernel size is an important hyperparameter. Table 5b summarizes the results for different convolutional kernel size choices for convolutional self-attention.

### 5.5. Equivariance Error Analysis

For a feature map $f(\cdot)$, the equivariance error is defined as

$$\mathcal{E}_{\text{eq}} = \|f(g \cdot x) - g \cdot f(x)\|_1, \qquad (23)$$

*Table 6.* Mean Equivariance Errors after *Lifting* layer and after *G-CSA* transformer blocks (before classification). Pre-classification equivariance errors for G-CNNs and non-equivariant vanilla ViTs are also provided for reference

| Dataset | RotMNIST | | CIFAR-10 | PatchCamelyon | |
|---|---|---|---|---|---|
| Group | *p4* | *p8* | *p4m* | *p4* | *p4m* |
| Lifting | 0 | 1.3e-3 | 0 | 0 | 0 |
| G-CSA | 1.5e-5 | 1.8e-2 | 1.7e-5 | 4.4e-5 | 2.8e-5 |
| G-CNN | 1.2e-5 | 2.0e-2 | 1.7e-5 | 2.7e-5 | 2.3e-5 |
| ViT | 3.1e-1 | 4.4e-1 | 3.7e-1 | 5.5e-1 | 5.2e-1 |

where $\|\cdot\|_1$ denotes the mean absolute difference over all spatial locations, channels, and batch elements. Table 6 shows equivariance error after the lifting layer and at the final feature after all G-CSA transformer blocks. Our models exhibit near-zero equivariance error across most groups. For the $p8$ group, errors accumulate starting at the lifting layer, which we attribute to interpolation artifacts introduced by rotations at angles such as $45°$ and $135°$, where pixel locations fall between the original grid. This results in approximation error at the input level. A more detailed equivariance analysis is provided in *Appendix* A.

### 5.6. Scaling REViT to ImageNet-1K

To demonstrate both the scalability of REViT and its potential utility as a roto-reflection equivariant backbone for downstream vision tasks, we additionally trained REViT on ImageNet-1K at a resolution of $224\times224$. Our experiments show that REViT can be reliably scaled to larger and more challenging datasets. Table 7 compares the performance of a $p4m$-equivariant REViT with two baselines: a vanilla ViT trained with rotation and flip augmentations and a $p4m$-equivariant RE-ResNet model. All models were trained for 300 epochs from scratch without pretrained weights or distillation. The results show that REViT consistently outperforms both the ViT-Small and equivariant ResNet baselines. Additional details on ImageNet REViT implementation, training procedure, and equivariance analysis are provided in *Appendix* B.

*Table 7.* Performance Comparison on ImageNet-1K.

| Model | Top-1 (%) | Top-5 (%) | Params. |
|---|---|---|---|
| ViT-S w/ aug | 72.08 | 89.54 | 22 M |
| RE-ResNet | 77.37 | 93.74 | 11 M |
| REViT | **79.27** | **94.45** | 18 M |

## 6. Discussion

Our results show the effectiveness of the proposed REViT with G-CSA as a roto-translation equivariant model. The experiments with three different datasets show that our simpler approach competes well with previous approaches based on augmenting self-attention with equivariance-preserving relative position encodings. We demonstrate that by exploiting the architectural components without position encoding like convolutional patch embeddings and convolutional projections for attention, roto-translation group equivariance can be achieved. We do that by lifting the input directly to $\mathbb{Z}^2 \times \mathbb{C}_N$ and then applying group convolutional self-attention.

Our results show that despite the simpler formulation of REViTs with G-CSA, in terms of model parameters and computations, we were able to achieve competitive results in comparison to typical group self-attention with RPE. Furthermore, across all the tested datasets and symmetry groups, REViTs show improvements over the corresponding equivariant G-CNN baselines without any significant increase in model sizes. In particular, our results on PatchCamelyon dataset show the effectiveness of our approach on larger image sizes for a real-world application that may benefit from roto-reflection equivariant classification. In this case, G-CSA not only outperforms existing approaches in terms of performance metrics, but it also does so with a simpler architecture and smaller roto-translation group sizes.

However, similarly to existing approaches for discrete group equivariance, a key limitation of the proposed group-equivariant architecture is that its computational and memory complexity still scales with the cardinality of the underlying symmetry group. Although the model consistently outperforms both prior equivariant approaches and comparably sized non-equivariant baselines, these gains come at the cost of increased inference latency and reduced computational efficiency. In particular, the repeated transformations and feature aggregation across group elements introduce substantial overhead relative to standard vanilla architectures of similar parameter count. As a result, while the method is effective in leveraging structured symmetries for improved predictive performance and generalization, its practical deployment in latency-sensitive or resource-constrained settings remains challenging.

## 7. Conclusion and Future Works

In this paper, we proposed REViTs with convolutional self-attention for roto-translation equivariant transformers. Our results demonstrate that our proposed approach outperforms the existing approaches for rotation and roto-reflection group equivariant image classification tasks. We also demonstrate that, unlike existing approaches, REViT can be scaled to larger and more difficult datasets like ImageNet (Deng et al., 2009). In the future, we plan to scale up our G-CSA-based REViT to larger ViT sizes and datasets with even higher resolutions. Furthermore, we also plan to demonstrate ImageNet-pretrained REViT's potential as group equivariant backbones for downstream tasks like oriented object detection and image segmentation.

## Impact Statement

This paper presents work on transformation equivariant transformers, and the goal of our work is to advance the field of Machine Learning. There are many potential societal consequences of our work, most of which we feel are applicable to all works aimed at advancing the field of computer vision and do not need to be specifically highlighted here. However, there are some application specific downstream impacts of equivariant vision models in areas such as medical imaging, robotics, and autonomous systems, where equivariance/invariance to geometric transformations is critical. By incorporating known structural symmetries directly into model design, these approaches can reduce reliance on large labeled datasets and extensive data augmentation, offering both scientific and engineering benefits. However, their increased computational cost also highlights the need for more efficient and scalable implementations to ensure practical and sustainable deployment.

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

# A. Equivariance Analysis of G-CSA

Given an input image $x$ and a discrete group element $g$, we compare model responses on the transformed input $g \cdot x$ with appropriately transformed representations of the original input. For each dataset and each group, $G$, we sample a fixed number of test examples (1,024) and evaluate them under all non-identity elements of the target symmetry group. For each batch (batch size 16) of inputs $x$, transformed inputs $g \cdot x$ are generated by applying the corresponding rotation and, when applicable, reflection in pixel space. Identity transformations are excluded from evaluation since they yield zero error by construction.

For each non-identity group element, two forward passes are performed: one on the original input $x$ and one on the transformed input $g \cdot x$. Intermediate representations are extracted at predefined stages of the network, and group actions are applied directly in feature space when required.

## A.1. Evaluation Metrics

### A.1.1. EQUIVARIANCE ERROR

In order to evaluate equivariance of feature representations, we measure the discrepancy between the representation obtained from a transformed input and the group-transformed representation of the original input. For a feature map $f(\cdot)$, the equivariance error is defined as

$$\mathcal{E}_{\text{eq}} = \|f(g \cdot x) - g \cdot f(x)\|_1, \tag{24}$$

where $\|\cdot\|_1$ denotes the mean absolute difference over all spatial locations, channels, and batch elements. This metric is evaluated at two stages: 1) after the lifting layer (*lifting equivariance error*) and 2) at the final feature representation after all the G-CSA transformer blocks and immediately preceding the classification head (*pre-class equivariance error*).

### A.1.2. PREDICTION CONSISTENCY

Since the final model predictions are expected to be invariant under the action of the group, we additionally evaluate invariance at the output level. Given logits $z(x)$ and $z(g \cdot x)$, we compute *prediction consistency*, defined as the fraction of samples for which $\arg \max z(g \cdot x) = \arg \max z(x)$.

## A.2. Evaluation Results

Table 8 reports the equivariance performance of REViTs across all three datasets and symmetry groups. Across all datasets, REViTs demonstrate near-perfect equivariance for smaller symmetry groups such as $p4$ and $p4m$. In these cases, the lift error is exactly zero and the pre-classification error is on the order of $10^{-5}$, with consistency reaching $100\%$. This indicates that intermediate feature representations transform almost exactly according to the prescribed group actions. As the group complexity increases (e.g., $p8$ and $p12$), small but measurable deviations from perfect equivariance emerge. These deviations are reflected in modest increases in both lift and pre-classification errors, along with slight reductions in consistency. Despite this, consistency remains above $98\%$ in all cases, suggesting that equivariance is largely preserved even for higher-order groups.

**RotMNIST.** For RotMNIST, REViTs achieve exact equivariance under the $p4$ group. For $p8$ and $p12$, lift and pre-classification errors increase slightly, but consistency remains high. This indicates that the model effectively captures rotational symmetries, even when the number of discrete rotations increases.

**CIFAR-10.** On CIFAR-10, the $p4$ and $p4m$ groups again yield exact or near-exact equivariance, with zero lift error and negligible pre-classification error. The $p8$ configuration results in higher errors, particularly at the pre-classification stage, although consistency remains at $100\%$. This suggests that while numerical deviations increase, they do not significantly impact output consistency for the tested samples.

**PCam** For PatchCamelyon, REViTs maintain perfect equivariance for $p4$ and $p4m$. The $p8$ group introduces larger lift and pre-classification errors, along with a small drop in consistency and increased variance.

**PCam (ds=2).** Since PatchCamelyon images are significantly larger, we also trained models with a modified lifting layer. This modified lifting made use of strided group equivariant convolutions to downsize as well as lift feature maps

*Table 8.* Equivariance evaluation for REViTs.

| Dataset | Group | Lifting Equivariance Err.↓ | Pre-Class Equivariance Err.↓ | Pred. Consistency (%)↑ |
|---|---|---|---|---|
| RotMNIST | $p4$ | $0.000000 \pm 0.000000$ | $0.000015 \pm 0.000002$ | $100.00 \pm 0.00$ |
| | $p8$ | $0.001272 \pm 0.001102$ | $0.018641 \pm 0.016121$ | $99.37 \pm 0.39$ |
| | $p12$ | $0.001326 \pm 0.000812$ | $0.016559 \pm 0.010130$ | $99.82 \pm 0.50$ |
| CIFAR-10 | $p4$ | $0.000000 \pm 0.000000$ | $0.000019 \pm 0.000001$ | $100.00 \pm 0.00$ |
| | $p4m$ | $0.000000 \pm 0.000000$ | $0.000017 \pm 0.000002$ | $100.00 \pm 0.00$ |
| | $p8$ | $0.002852 \pm 0.002472$ | $0.043629 \pm 0.037864$ | $100.00 \pm 0.00$ |
| PCam | $p4$ | $0.000000 \pm 0.000000$ | $0.000044 \pm 0.000024$ | $100.00 \pm 0.00$ |
| | $p4m$ | $0.000000 \pm 0.000000$ | $0.000028 \pm 0.000015$ | $100.00 \pm 0.00$ |
| | $p8$ | $0.003512 \pm 0.003245$ | $0.040425 \pm 0.035378$ | $99.72 \pm 2.63$ |
| PCam (ds=2) | $p4$ | $0.029141 \pm 0.004238$ | $0.042795 \pm 0.005889$ | $99.45 \pm 1.78$ |
| | $p4m$ | $0.018769 \pm 0.008292$ | $0.031050 \pm 0.013799$ | $99.50 \pm 1.89$ |
| | $p8$ | $0.017205 \pm 0.003770$ | $0.065345 \pm 0.029455$ | $99.86 \pm 0.92$ |

*Table 9.* Equivariance evaluation for Vanilla ViTs.

| Dataset | Group | Pre-Class Equivariance Error↓ | Prediction Consistency (%)↑ |
|---|---|---|---|
| RotMNIST | $p4$ | $0.306930 \pm 0.007311$ | $84.34 \pm 4.82$ |
| | $p8$ | $0.440506 \pm 0.115933$ | $80.13 \pm 6.22$ |
| | $p12$ | $0.463208 \pm 0.096067$ | $79.27 \pm 6.01$ |
| CIFAR-10 | $p4$ | $0.412258 \pm 0.031058$ | $86.95 \pm 3.89$ |
| | $p4m$ | $0.376030 \pm 0.041067$ | $88.21 \pm 4.22$ |
| | $p8$ | $0.552319 \pm 0.123461$ | $81.39 \pm 7.04$ |
| PCam | $p4$ | $0.548849 \pm 0.062911$ | $94.24 \pm 6.05$ |
| | $p4m$ | $0.521713 \pm 0.065477$ | $94.25 \pm 6.05$ |
| | $p8$ | $0.574886 \pm 0.055969$ | $91.77 \pm 7.08$ |

before sending them to G-CSA transformer blocks. PCam (ds=2), i.e., downsizing factor of 2, here refers to such a model configuration. The PCam models with stronger downsizing present a more challenging scenario. In this case, non-zero lift and pre-classification errors appear even for $p4$ and $p4m$, although consistency remains close to $100\%$. The $p8$ group achieves comparable lift error to $p4m$ but exhibits the largest pre-classification error, suggesting that aggressive downsizing amplifies equivariance imperfections, particularly in later network stages.

Overall, these results demonstrate that REViTs exhibit strong equivariance properties across datasets and symmetry groups. While performance degrades slightly with increasing group complexity and data difficulty, deviations from ideal equivariance remain small and rarely affect output consistency.

### A.2.1. COMPARISON WITH G-CNN AND VIT

We conducted the exact same equivariance analysis on vanilla VITs with typical self-attention (Dosovitskiy et al., 2021) and G-CNNs as well. The results of those analyses are summarized in Table 9 and Table 10, respectively. The equivariance errors and prediction consistencies for ViTs are significantly worse. This is because the ViTs neither have any equivariant architectural components to preserve equivariance in feature maps, nor are they trained here using data augmentations which are generally used to produce transformation invariant classification behavior, i.e. high prediction consistency. The G-CNNs, on the other hand, display almost perfect prediction consistency and negligible equivariance error just like REViTs. Furthermore, both G-CNNs and REViTs display slight but similar erratic behavior for higher-order rotation groups, like $p8$ and $p12$. We elaborate on it further in the following subsection.

### A.2.2. ON INTERPOLATION ARTIFACTS AND LIFTING ERROR

In Section 5.5, we briefly mention interpolation artifacts as the likely cause of lifting errors for groups with higher rotation orders. Here, with the help of Figure 4, we elaborate how this error starts accumulating at the input stage of the test even before the transformed image reaches G-CSA.

*Table 10.* Equivariance evaluation for G-CNNs

| Dataset | Group | Pre-Class Equivariance Error↓ | Prediction Consistency (%) ↑ |
|---------|-------|-------------------------------|------------------------------|
| RotMNIST | $p4$ | 0.000012±0.000001 | 100.00±0.00% |
| | $p8$ | 0.020938±0.018117 | 100.00±0.00% |
| | $p12$ | 0.026480±0.016213 | 100.00±0.00% |
| CIFAR-10 | $p4$ | 0.000023±0.000001 | 100.00±0.00% |
| | $p4m$ | 0.000017±0.000001 | 100.00±0.00% |
| | $p8$ | 0.022805±0.019764 | 94.70±5.15% |
| PCam | $p4$ | 0.000027±0.000002 | 100.00±0.00% |
| | $p4m$ | 0.000023±0.000002 | 100.00±0.00% |
| | $p8$ | 0.027088±0.024306 | 100.00±0.00% |

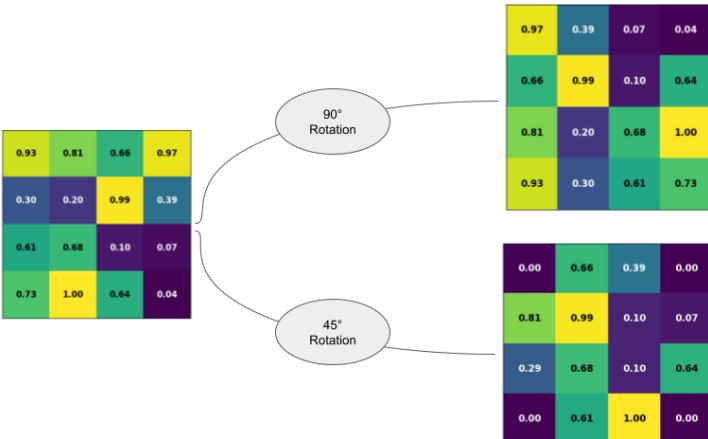

*Figure 4.* Illustration of interpolation-induced approximation error under discrete rotations. Left: original discrete image on a pixel grid. Top-right: rotation by $90°$, which aligns exactly with the grid and preserves values exactly. Bottom-right: rotation by $45°$, where pixel locations fall between grid points and bilinear interpolation is required, resulting in mixed values and approximation artifacts. This discrepancy explains the observed lifting equivariance errors for rotation groups containing non-grid-aligned transformations (e.g., $p8$).

Let $f : \mathbb{Z}^2 \rightarrow \mathbb{R}$ denote a discrete image, and let $R_\theta$ be a rotation operator by angle $\theta$. For grid-aligned rotations such as $\theta = 90°$, the transformed coordinates $R_\theta x \in \mathbb{Z}^2$ remain on the integer lattice:

$$(f \circ R_\theta)(x) = f(R_\theta x), \tag{25}$$

which corresponds to a permutation of pixel values and preserves equivariance exactly.

In contrast, for non-grid-aligned rotations such as $\theta = 45°$, the transformed coordinates do not satisfy $R_\theta x \in \mathbb{Z}^2$. The rotated output must therefore be resampled via an interpolation operator $\mathcal{I}$:

$$(f \circ R_\theta)(x) \approx \mathcal{I}\big(f(R_\theta x)\big). \tag{26}$$

This interpolation introduces a discrepancy between the ideal continuous transformation and its discrete realization, producing approximation error at the input or lifting stage. As a result, equivariance violations accumulate for rotation groups containing such angles, even when the network itself is theoretically equivariant.

### A.3. Generalization to Random Transformations

We evaluate the generalization ability of REViTs beyond the transformation groups they are trained for by applying random transformations to test images and measuring the stability of model predictions. For each input sample, we generate multiple transformed versions using random rotation angles uniformly sampled from $0°$ to $360°$, combined with random flips with a probability of 50%. Because these transformations lie outside the discrete symmetry groups for which the equivariant

*Table 11.* REViT's generalization to random transformations in comparison to vanilla ViTs using SA

| Model / Group | Avg. Prediction Consistency (%)↑ | Avg. Probability Difference↓ |
|---|---|---|
| REViT ($p4$-CSA) | 97.19 | 0.001144 |
| REViT ($p8$-CSA) | 96.11 | 0.000084 |
| REViT ($p4m$-CSA) | 97.73 | 0.000444 |
| REViT ($p12$-CSA) | 99.87 | 0.000013 |
| **OVERALL (REViT)** | **96.99** | **0.000510** |
| ViT (SA) | 85.38 | 0.010658 |
| **Difference (REViT – ViT)** | **+11.61** | **-0.010148** |

models are designed, exact equivariance cannot be enforced. Instead, we assess invariance by comparing the original and transformed outputs using prediction consistency and changes in softmax probabilities. This evaluation is performed across all three datasets and compares REViT models against a non-equivariant Vanilla ViT baseline, thereby quantifying how well discrete equivariance generalizes to unseen continuous transformations. Table 11 summarizes the results from this comparison and shows how well REViTs generalize to out-of-group transformations. REViTs have significantly higher prediction consistencies and lower logit probability differences compared to typical ViTs.

# B. REViT on ImageNet

## B.1. Implementation

The implementation of REViT on ImageNet is different from the other REViTs in the following two ways.

### B.1.1. LIFTING STEM

Instead of a single lifting layer, ImageNet REViT has a deeper, hierarchical lifting stem that performs feature extraction and spatial reduction progressively before the main attention layers. Conceptually, this changes the front end from a minimal projection layer into a structured lifting encoder. This builds stronger low-level equivariant features earlier, reduces spatial resolution more deliberately, and presents the transformer with a more compact and semantically processed representation. This makes the model behave less like a flat transformer applied directly after lifting and more like a hybrid hierarchical vision backbone.

### B.1.2. WINDOWED G-CSA IMPLEMENTATION

In smaller REViTs, each G-CSA block operates over the full spatial extent of the feature map, allowing immediate long-range interaction but making the computational and memory cost grow quickly with image size. However, a windowed G-CSA restricts attention to local windows and relies on the hierarchical architecture to expand the effective receptive field over depth and across stages.

## B.2. Training

REViT was trained on ImageNet using distributed data parallelism on four Nvidia GeForce RTX4090 GPUs, with a per-GPU batch size of 128, giving an effective batch size of 512. We optimize the model for 300 epochs with AdamW, using an initial learning rate of 3e-4, weight decay of 0.05, and a learning-rate schedule consisting of 20 epochs of linear warmup followed by cosine decay. The model is instantiated with a window size of 7, and a 3x3 equivariant kernel for the Q/K/V projections in the attention layers.

## B.3. Equivariance Analysis

Since ImageNet REViTs were implemented differently, we also evaluated the architectural equivariance of those REViTs on the ImageNet validation set using (24). In this setting, no trained checkpoint is loaded; the model is randomly initialized, so the experiment isolates equivariance induced by the network design rather than invariance learned through optimization. 1024 images were taken from ImageNet validation set and processed with standard ImageNet evaluation transforms: resize to 256, center crop to 224 x 224, and normalization with ImageNet mean and standard deviation.

*Table 12.* Equivariance evaluation for ImageNet REViTs with windowed G-CSA.

| Group | Lifting Stem Equivariance Err.↓ | Pre-Class Equivariance Err.↓ |
|-------|--------------------------------|------------------------------|
| $p4$ | $0.000142 \pm 0.000017$ | $0.00218 \pm 0.000471$ |
| $p4m$ | $0.001316 \pm 0.000614$ | $0.00008 \pm 0.000034$ |
| $p8$ | $0.001474 \pm 0.000366$ | $0.000102 \pm 0.000013$ |

For each sampled image and each non-identity element of the $p4$, $p4m$, and $p8$ groups, we compared the network response to the transformed input against the transformed response to the original input. Equivariance was measured at the stem output and immediately before the classification head (after all G-CSA blocks). Table 12 summarizes the results of these equivariance tests.

