# OpenReview forum: "REViT: Roto-reflection Equivariant Convolutional Vision Transformer"
_ICML.cc/2026/Conference — ICML 2026 regular_

### Official Review · Reviewer_pJFj · 2026-02-20

**Soundness:** 3
**Presentation:** 3
**Significance:** 3
**Originality:** 3
**Overall Recommendation:** 5
**Confidence:** 4

**Summary:**

This work proposes a discrete roto-reflection group equivariant vision transformer using convolutional attention. It removes the positional encoding from a transformer, and instead uses a group convolutional self-attention to ensure the model is equivariant as well as captures local information. Additionally, there is a lifting layer to project the input to a group space with appropriate dimensions before applying the convolutional attention. Experimental results on rotated MNIST, PatchCamelyon, and CIFAR10 show improvement over prior works with same number of parameters.

**Compliance With Llm Reviewing Policy:**

Affirmed.

**Final Justification:**

The initial draft had very limited experiments making it difficult to verify if the method works in larger scale datasets. However, additional experiments from the author convinced that the method is scalable and hence a good contribution

**Key Questions For Authors:**

Please address the weaknesses

**Limitations:**

yes

**Strengths And Weaknesses:**

Strengths:
- The paper is well written and proposes a simple roto-reflection equivariant transformer architecture for vision tasks.
- The paper designs a provably equivariant transformer model by removing positional encodings and using group convolutional layers
- Prior works such as Romero et al. 2021 have shown that training equivariant attention networks can be unstable. Hence, the proposed architecture that achieves good performance is a positive contribution.


Weaknesses:
- The experiments section is very weak with only limited applications on rot-MNIST, PatchCamelyon, and CIFAR-10. I think the authors should provide more applications as use cases including Imagenet but not limited to it. Otherwise, the overall contribution of this work looks limited.
- The novelty of the work is limited. As already mentioned in the paper, there are prior works with equivariant attention mechanism, lifting layers, as well as convolutional attention networks. Even though, the paper puts these components together, the overall novelty is limited.

---

> ### Author Rebuttal · Authors · 2026-03-31
>
> We thank the reviewer for seeing the strengths of our work and providing comments about how it could be improved. The following is our response to those comments.
>
> ***Experiments on larger datasets, e.g. ImageNet***
> We also conducted experiments on Imagenet-1K. However, due to time and infrastructure costs, we could not conduct thorough experiments and ablations on ImageNet at the time of submission. Thus, we listed ImageNet implementation as our future work.
> However, since then, we have conducted further experiments on ImageNet REViTs and the following table summarizes REVIT's performance. All models in this table were trained from scratch on ImageNet-1K with the same training configurations and without any pretrained weights or distillation.
>
> | Model | Top-1 | Top-5 | Params |
> | :--- | :--- | :--- | :--- |
> | ViT-S | 0.703 | 0.895 | 22M |
> | RE-Resnet | 0.701 | 0.895 | 12M |
> | REViT-S | ***0.705*** | ***0.897*** | 16M |
> | REViT-VS | 0.692 | 0.891 | 7M |
> | REViT-T | 0.662 | 0.873 | 3M |
>
> All the results reported in this table, except for REViT-T, are on models trained for 150 epochs using the same optimizer and learning rate schedulers. ReViT-T was trained for 200 epochs.
>
> REViT's for ImageNet slightly differ from the models for the other datasets: the lifting layer is implemented in conjunction with  a convolutional stem  that downsizes the feature by a factor of 4. Downsampling is performed by group-equivariant blocks to ensure that the equivariance is preserved.
>
> We also plan to summarize the ImageNet-1K results in the main paper and explain the implementation details in the appendix.
>
>
> ***On Novelty and Contribution***
> We agree that the individual components (equivariant attention, lifting layers, convolutional attention) are not novel in isolation. Our contribution lies in how they are combined into G-CSA, a stable and effective architecture for 2D image classification, which addresses a known limitation in prior work.
>
> In particular, as also noted in your review, previous studies (e.g., Romero et al., 2021) report training difficulties with equivariant attention networks. A key contribution of our work is showing that these components can be integrated in a way that trains reliably while achieving strong performance.
>
> Additionally, we contribute a systematic equivariance error analysis, which is not provided in the baseline methods we compare against. This offers new empirical insight into how architectural choices affect equivariance in practice.

---

> > ### Author Rebuttal · Reviewer_pJFj · 2026-04-04
> >
> > The authors have provided experiments on imagenet showing the effectiveness of the architecture. Further, even though the novelty is limited, the engineering aspect of the work to stabilize the equivariant attention is good. I am updating my score to 5.

---

> > > ### Author Response · Authors · 2026-04-06
> > >
> > > Thanks for engaging with our rebuttal comments, and we are glad that our rebuttal adequately addressed your concerns. We also really appreciate the raised score. Thank you.

---

### Official Review · Reviewer_tcpL · 2026-03-01

**Soundness:** 3
**Presentation:** 2
**Significance:** 3
**Originality:** 3
**Overall Recommendation:** 5
**Confidence:** 4

**Summary:**

The paper proposes a rotation and reflection equivariant vision transformer model, where the two main components are the equivariant lifting of the input image and the convolution self attention to give the model equivariance throughout the architecture. Lifting to the group is done in the 'standard' way by rotating the filters to obtain rotated feature maps, similar to GCNN. The main contribution of the paper lies in the way that REViT handles the self attention mechanism. The Key, Query and Values are projected into two-dimensional spatial projection and 1-dimension that corresponds to the order of the group that REViT is being lifted to. This effectively mirrors the MHSA mechanism such that it matches the group transformation of the lifting layer. The author show both analytically (in section 4.2.3) and empirically (in appendix A.1.1) that the proposed method does indeed achieve equivariance to the desired transformations. Furthermore, this is supported by the evaluation on CIFAR-10, rotated MNIST, and PatchCamelyon, where REViT generally outperforms baseline models.

**Compliance With Llm Reviewing Policy:**

Affirmed.

**Final Justification:**

The rebuttal addressed most of my concerns, with only small presentational issues remaining. Please refer to my rebuttal acknowledgment for detailed explanation.

**Key Questions For Authors:**

Please refer to the strength and weakness section, mainly:
1. How does the proposed model perform on larger and higher resolution datasets
2. How does the model compare to data augmentation schemes, both in terms of accuracy and training time
2. How doe the computation time compare to baseline methods when parameter counts are similar
3. Greater detail in related works section

**Limitations:**

No limitations section. While it is acceptable by ICML standard, the standard impact statement is disappointing. Improved classification and CV architecture have many potential negative impacts, such as surveillance. I fact right now Anthropic is taking a stand on using AI models for weapons and surveillance tasks.

**Strengths And Weaknesses:**

I gave the paper a weak reject. However I believe that there is significant merit to the paper and the score should be raised to a 5 if the authors are able to address the concerns during the rebuttal period. **Edit**: after significan soul searching (and reviewing the other papers ICML tortured me with) I’ve raised my original assessment to 4. Everything else remains relevant tho…

On **originality** I scored the paper 3. The idea of designing for group symmetries is not new in computer vision, which the authors have noted both in their introduction and related works. Furthermore, the concept of convolutional self attention has also been proposed. The originality of REViT is from its efficient implementation that both shows equivariance empirically and analytically.

On **significance**  I scored the paper 3. Designing for more computation efficient neural network architectures is increasing beneficial in the face of rising cost of storage and computation. When the group transformation is well known, such as the case of rotation in images, constraining the network to respect these symmetries is important. However, the significance is not higher as the authors have only demonstrated REViT with relatively limited evaluation (for reference see other CV paper's evaluation scope [1], **this isn't a push to cite this paper** lol, it isn't mine but just a good example of the scope of evaluation).

On **soundness** I scored the paper 2. The experimental set up is fine and sound, however the limited evaluation is still a concern to me. While the authors acknowledged that scaling up to large scale datasets such as image net is necessary, this should not be a limitation to the paper - it should be included. As of now the evaluated datasets are kinda small and low resolution. Additionally I believe computation time should be reported. Even though the parameter count may be similar, due to the need to compute attention across the group I believe that the computational time would increase proportionally (as discussed in the original GCNN paper). Furthermore, the baselines a little weak, considering there is significant development after GCNN.

On **presentation** I scored the paper 2. The overall paper reads fine, however more details should be added to the methods section. IMO the proof on equivariance can be abbreviated (with details moved to the appendix) and the equivariance error moved into the main body. Furthermore, the related works seems rushed and incomplete. Specifically rotation convolution and equivariant pooling should be combined. The section of group convolution is missing a lot of related work in general [2,3,4], as well as more specialized such as those studying 3D rotations [5,6,7,8], perceptual variations [9,10], and general groups [12,13].

[1] Mounir Messaoudi, et al. "Designing Affine-Invariant Neural Networks for Photometric Corruption Robustness and Generalization." ICLR, 2026.\
[2] Risi Kondor, and Shubhendu Trivedi. "On the generalization of equivariance and convolution in neural networks to the action of compact groups." ICML, 2018.\
[3] Taco Cohen, Mario Geiger, and Maurice Weiler. "A general theory of equivariant cnns on homogeneous spaces." NeurIPS, 2019.\
[4] Daniel Worrall, et al. "Harmonic networks: Deep Translation and Rotation Equivariance." CVPR, 2017.\
[5] Carlos Esteves, et al. "Learning so (3) Equivariant Representations with Spherical CNNs." ECCV, 2018.\
[6] Taco Cohen, et al. "Spherical CNNs." ICLR, 2018.\
[7] Simon Batzner, et al. "E (3)-equivariant graph neural networks for data-efficient and accurate interatomic potentials." Nature Comm. 2022.\
[8] Carlos Esteves, et al. "Polar transformer networks." ICLR, 2018.\
[9] Attila Lengyel, et al. "Color equivariant convolutional networks." NeurIPS, 2023.\
[10] Yulong Yang, Felix O'Mahony, and Christine Allen-Blanchette. "Learning color equivariant representations." ICLR, 2025.\
[12] Marc Finzi, et al. "Generalizing convolutional neural networks for equivariance to lie groups on arbitrary continuous data." ICML, 2020.\
[13] Taco Cohen, et al. "Gauge equivariant convolutional networks and the icosahedral CNN." ICML, 2019.

---

> ### Author Rebuttal · Authors · 2026-03-31
>
> We thank the reviewer for a thorough and insightful review of our work. The following is our response to the questions raised by the reviewer.
>
> ***On Larger Dataset: ImageNet***
> We also conducted experiments on ImageNet-1K. Our models show competitive performance on ImageNet as well. Please refer to our response to ``Reviewer ***pJFj***" for more details about REViTs on ImageNet.
>
>
> ***On Comparison Data Augmentation Schemes***
> REViTs always outperformed similarly sized and trained non-equivariant models. The performance of non-equivariant transformer models when trained with rotation and flip augmentations improves significantly. However, when controlled for model size and number of epochs, that improved performance still lags behind REViT. The numerical performance difference is described in Table 4 in the paper and our response to ``Reviewer ***wevz***".
>
> However, the comparison of total training time is a bit more nuanced. In terms of raining times per epoch, $p4$ REViT and $p4m,p8$ REVIT are on average  2.67x and 3.64x times, respectively, slower than those of standard ViTs with rotation and flip data augmentation. Although the time per epoch is higher, ReViT typically exhibits much better sample efficiency. For datasets where orientation really  matters, i.e., PatchCamelyon, a model can often reach its peak accuracy in 30-45\% fewer epochs than a baseline ViT.
>
>
> ***On Computational times*** REViTs are significantly faster than equivariant transformer baselines but slower than G-CNNs and vanilla ViTs.
>
> In the paper, we opted to present the complexity comparison in terms of memory requirement and number of operations as those metrics are not as hardware dependent as inference time. That said, the following is the inference time comparison on the hardware we tested on:
>
> The average inference runtimes on an RTX4090 GPU for a batch of 32
> 96x96 images (PCam) were $\approx$12 ms for REViT($p4$), $\approx$40 ms for SA with RPE ($p4$), and $\approx$6 ms for G-CNNs ($p4$) of similar sizes of
> $\approx$100K parameters. However, for vanilla ViT, the inference time was $\approx$2ms. G-CNNS were implemented entirely with the 'escnn' python package which is well optimized for fiber group operations.  However, REViTs were only implemented partially with 'escnn' which may explain the difference in implementation efficiency.
>
>
> ***Greater Detail in related works***
> Thank you for the detailed feedback. We would like to clarify that the scope of our related work section was intentionally focused on 2D group equivariance in the context of image classification, which is the setting our method is designed for and evaluated on. As such, we prioritized works directly relevant to planar transformations (e.g., rotations and translations in 2D) and their practical implementations in CNN-based architectures. This design choice explains why broader formulations—such as general compact groups, Lie groups, or 3D equivariant models—were not covered exhaustively in the current version.
>
> That said, we agree that the paper would benefit from a more comprehensive discussion that better situates our work within the larger landscape of equivariant learning. In particular, the references spanning general group convolution theory, 3D equivariance (e.g., SO(3), E(3)), perceptual symmetries such as color, and gauge/Lie group formulations—are highly relevant from a conceptual standpoint, even if they extend beyond our immediate problem setting. We will incorporate these works in the revision and explicitly clarify the distinction between our targeted 2D setting and these broader frameworks.
>
> Regarding the related work [1] mentioned by the reviewer, we agree that they present some useful analyses that we can utilize for our models too. However, since it is very recent (ICLR-2026), we were not aware of it during the writing of this paper. Additionally, in Appendix 1.3, we did evaluate REViT on robustness to transformations for which it was not specifically trained.
>
> ***On Impact Statement***
> We agree with the reviewer's opinion on the societal impact of our work. We opted for the standard Impact Statement allowed by ICML because because we thought that those implications are not particularly specific to our work. However, since Impact Statement does not count towards the page limit of the paper, we will expand further on those impacts in the revised version of the paper.
>
> $\dagger$ Since ICML allows for an extra page for the final paper, if accepted, we believe we can easily incorporate all of the aforementioned revisions into the paper.

---

> > ### Author Rebuttal · Reviewer_tcpL · 2026-04-01
> >
> > Thank you for the detailed rebuttal. I believe that most of my concern has been addresses, and I intent to raise my score to 5.
> >
> > I believe that the model not only demonstrates strong performance of rotation transformations, but also potentially serves as a backbone for future implementation of other group equivariant ViTs.
> >
> > I do have a few follow up questions. Unfortunately I don't believe that addressing these will raise the score to a 6, but for scientific completeness I hope the authors address them.
> >
> > > On Larger Dataset: ImageNet
> >
> > The results seem fine - I don't expect much rotation distribution shift between train and test on the ImageNet dataset. If the authors have the trained model checkpoints can they check how the models perform on rotated-ImageNet-1K> I would suspect much better separation compare to vanilla results.
> >
> > Just as a side note since I'm not really suppose to ask for additional exp during the discussion phase, the results of this will not impact my evaluation.
> >
> > > On Comparison Data Augmentation Schemes
> >
> > My bad for missing the obvious table in the original submission. Thank you for the addition CvT experiment. These results significantly strength the submission.
> >
> > > On Computational times
> >
> > Thank you for the compute comparison. A more detailed comparison/table should be included. The results show that ReViT does indeed require more compute, but that is expected. These tradeoff should be made explicit in the paper.
> >
> > Additionally, can the authors also include the memory value for the inference on the same hardware? Sometime the parameter value does not necessarily scale linearly to memory due to the need to store filters on an orbit. Including both of these will make the analysis more stringent.
> >
> > > Greater Detail in related works
> >
> > Sorry about the confusion on [1] - this was simply an paper in my recent memory where I thought they have great evaluation. Obviously the authors do not need to disucss/include concurrent work as defined by ICML.
> >
> > Furthermore the robustness to out of distribution transformations is an interesting observation - I assume this is due to both the inductive bias as well as the expressive capacity of the baseline vision transformer model. Very cool.
> >
> > > Impact statement
> >
> > Thanks you. I know this is annoying but I do believe this is important.
> >
> > Edit: soundness goes from 2 -> 3 as well.

---

> > > ### Author Response · Authors · 2026-04-07
> > >
> > > First of all, thank you for extremely helpful engagement with our work. We appreciate both your comments and the raised score.
> > >
> > > Following is our response to your follow-up questions.
> > >
> > > ***On Rotated ImageNet validation***
> > > We ran some quick tests by validating the ImageNet-1k trained models on a rotated  ImageNet-1k validation set. And as you suspected, the separation is much more noticeable.
> > > | Model | Top-1 | Top-5 |
> > > | :--- | :--- | :--- |
> > > | ViT | 0.521 | 0.745 |
> > > | REViT | 0.591 | 0.794 |
> > >
> > > These are preliminary results based on the best checkpoints saved. We plan to conduct  more comprehensive evaluations for the  updated version of the paper.
> > >
> > > ***On Computational times***
> > > We will add the reported times above along with further computational time analyses table. We will also explicitly explain the understandable but a significant computational time trade-off for performance improvement and equivariance.
> > >
> > > We will also add the memory column to the computation time/complexity analysis table. Based on our tests, your intuition is right, the same parameter count doesn't equate to same memory footprint. We observed that the groups with more group elements had a higher memory footprint even if the number of parameters are same. The following table gives a brief summary how the REViTs scale with group sizes.
> > >
> > > | Model | Group Elements | Memory Scale Factor (vs ViT) |
> > > | :--- | :--- | :--- |
> > > | REViT (Z2) | 1 | 1 |
> > > | REViT (p4) | 4 | 1.374 |
> > > | REViT (p4m, p8) | 8 | 1.821 |
> > >
> > > These numbers are averaged for models of similar size over several different inference batch sizes across two different datasets: CIFAR-10 and Rotated MNIST.
> > >
> > > *Memory Scale Factor* is defined as *TotalMemory_per_Image/num_of_parameters* divided by  Vanilla ViT's *TotalMemory_per_Image/num_of_parameters*
> > >
> > > Hopefully, these responses answer your follow up questions.
> > > Thank you again for helping us significantly improve the paper.

---

### Official Review · Reviewer_wevz · 2026-03-09

**Soundness:** 3
**Presentation:** 2
**Significance:** 2
**Originality:** 3
**Overall Recommendation:** 4
**Confidence:** 2

**Summary:**

This paper proposes REViT, a discrete roto-reflection equivariant vision transformer that removes explicit positional encoding and instead builds equivariance through convolutional patch embeddings, a lifting layer from the image grid to a group-indexed feature space, and a new group convolutional self-attention (G-CSA) module. The core idea is to lift features to $\mathbb{Z}^2 \times C_N$, compute equivariant $Q/K/V$ with group convolutions, and perform attention jointly over spatial neighborhoods and group elements; the paper also provides a proof sketch that G-CSA is group equivariant under its formulation. Empirically, the method is evaluated on Rotated MNIST, CIFAR-10, and PatchCamelyon, where it is compared against group-equivariant self-attention with relative positional encoding and against group-equivariant CNNs, indicating that REViT generally matches or outperforms the RPE-based equivariant transformer baselines while reducing operation counts and memory in the complexity comparison, and it also improves over the corresponding G-CNN baselines on the tested classification tasks.

**Compliance With Llm Reviewing Policy:**

Affirmed.

**Final Justification:**

The rebuttal satisfactorily answers my implementation-level questions on equivariant MLP/normalization and adds the requested matched CvT baseline, which strengthens the empirical case. So I raise my score to 4.

**Key Questions For Authors:**

1. How are normalization and MLP layers implemented to preserve equivariance? Specifically, are normalization statistics computed independently for each spatial location and group element, and are there any biases or affine parameters that could break equivariance?
2. For the G-SA baselines with RPE, could you provide complete training details and architecture parity information, including optimizer, schedule settings, augmentations, model depth/width/head configuration to ensure a fully fair comparison?
3. Could you compare against a matched non-equivariant CvT-style baseline with similar parameter budget?

**Limitations:**

yes

**Strengths And Weaknesses:**

Strength:

- Introduces a simple, modular G-CSA layer that combines CvT-style convolutional projections with group convolution and lifting, sidesteping relative positional encodings while enforcing discrete roto-(reflection) equivariance.
- Compares against strong and directly relevant baselines: group-equivariant self-attention with relative positional encoding (Romero & Cordonnier, 2021) and G-CNNs (Cohen & Welling, 2016; Romero et al., 2020)

Weaknesses:

1. The formal proof in Section 4.2.3 establishes equivariance only for G-CSA, while REViT is presented as a modified standard ViT architecture. The paper does not justify how equivariance is preserved through the rest of the block or through the final prediction head.
2. Experiments are confined to RotMNIST, CIFAR-10, and PatchCamelyon classification. The paper does not provide ImageNet-scale evidence, nor any downstream detection or segmentation results, even though those extensions are highlighted as motivating applications and future directions.
3. The setup section omits key training details needed to assess baseline parity and reproducibility, and several tables contain missing or incomplete entries. So it is difficult to judge whether the reported gains come from the proposed equivariant structure rather than undocumented differences.
4. REViT inherits convolutional patch embedding and convolutional attention ideas from CvT-style architectures, yet the ablation compares mainly against vanilla ViT-SA and prior equivariant models. A non-equivariant CvT-style baseline of matched size would better separate the effect of group equivariance from the effect of simply replacing vanilla attention with convolutional attention.

---

> ### Author Rebuttal · Authors · 2026-03-31
>
> We thank the reviewer for seeing the key contributions of our work and the resulting superior performance. We also appreciate their comments for improving the paper. The following is our response to the questions raised by the reviewer.
>
> ***On equivariance of MLP layers and normalizations.***
> The network keeps a fixed field: at every pixel the channel block is a vector that transforms under the discrete rotation group, not a free list of scalars.
>
> MLP: As in previous works on equivariant models, including [1], we implement the 'dense linear layers' in MLP blocks by using only 1×1 group-equivariant convolutions whose weights are constrained so that, for each spatial location, the map commutes with rotating that fiber.  We also turn bias off on those layers so that no unconstrained offset is added.
>
> Normalization: We use batch normalization that respects the fiber structure: moments are pooled over the batch and the image, and across the coordinates that rotate together within each group block, instead of treating every channel as an independent scalar with its own scale. Any learned scale/shift is tied to that block structure, not arbitrary per-channel affines.
>
> We only theoretically proved the equivariance of G-CSA because that is the main equivariant component that we proposed in this paper. Other components, besides, G-CSA, have previously been shown to be group-equivariant. Furthermore, the equivariance analysis of REViT shows that the equivariance is preserved throughout the network.
>
> ***On comparison with G-SA with RPE.***
> For all G-SA with RPE baselines, we used the model training configurations provided by [1] and their code. Since G-SA with RPE models are heavy and computationally expensive to scale, we used REViTs of similar or smaller sizes (Table 1 and Table 2 in the paper) to make a fair comparison. We used the same optimizer (AdamW) with the same cosine annealing learning rate scheduler with 10\% linear warmup, and trained for the same number of epochs: 300 for CIFAR-10 and 200 for Rotated MNIST and PatchCamelyon. We will add this and other implementation details to the paper appendix and if accepted, we plan to release the code too.
>
> ***On Comparison with CvT-style baseline.***
> REViTs do perform better than CvTs with similar parameter budget.
>
> In the paper, we opted to go with standard ViT as our non-equivariant baseline since it is more well known. However, both ViT and CvT showed very similar performances when compared with REViTs of the same sizes (depth, width, heads, and in the case of CvT, convolutional kernel sizes). The following table shows the accuracy comparison of CvT against REViT. CvTs were trained by replacing G-CSA in the corresponding REViTs with CSA and group equivariant 1x1 MLP convolutions 1x1 standard 2D convolutions. CvT were trained with rotation and flip augmentations.
>
> | Model | Rotated MNIST | CIFAR-10 |
> | :--- | :--- | :--- |
> | CvT | 91 | 83 |
> | REViT | 98 | 90 |
>
>
> ***Results on ImageNet***
> We also conducted experiments on Imagenet-1K. Our models showed competitive performance on ImageNet as well. Please refer to our response to ``Reviewer ***pJFj***" for more details about REViTs on ImageNet.
>
>
>
> [1] Romero, D. W. and Cordonnier, J.-B. Group equivariant stand-alone self-attention for vision. In International Conference on Learning Representations, 2021

---

> > ### Author Rebuttal · Reviewer_wevz · 2026-04-03
> >
> > The rebuttal satisfactorily answers my implementation-level questions on equivariant MLP/normalization and adds the requested matched CvT baseline, which strengthens the empirical case. However, the end-to-end equivariance justification beyond G-CSA remains only partially addressed, and baseline parity / reproducibility details are still incomplete. I will raise my score to 4.

---

> > > ### Author Response · Authors · 2026-04-06
> > >
> > > Thank you for engaging with our rebuttal comments, and we are glad that our rebuttal satisfactorily answered some of your concerns. We also appreciate the raised score.
> > >
> > > Furthermore, in the revised version of the paper, we will try to address your remaining concerns by expanding more on the equivariance of components beyond G-CSA and releasing the implementation code for REViTs and the G-CNNs and ViT baselines. The G-SA with RPE baselines were implemented using the code published by Romero et. al., 2021.

---

### Decision · Program_Chairs · 2026-04-30

**Decision:**

Accept (regular)

**Comment:**

Initial reviews were mixed. Reviewers valued the simple and effective method, the achievement of equivariant attention without positional encoding, and good results. Concerns were raised about novelty, the small scale of experiments, lacking details and baselines.

The rebuttal resolved most of reviewers concerns and included convincing larger-scale experiments on ImageNet. All reviewers increased their scores and recommended acceptance.

I follow the reviewer consensus and recommend acceptance.